



# Population exposure to outdoor NO$_2$, black carbon, particle mass, and number concentrations over Paris with multi-scale modelling down to the street scale

Soo-Jin Park[1], Lya Lugon[1], Oscar Jacquot[1], Youngseob Kim[1], Alexia Baudic[7], Barbara D'Anna[8], Ludovico Di Antonio[3], Claudia Di Biagio[4], Fabrice Dugay[7], Olivier Favez[5], Véronique Ghersi[7], Aline Gratien[4], Julien Kammer[8], Jean-Eudes Petit[6], Olivier Sanchez[7], Myrto Valari[2], Jérémy Vigneron[7], and Karine Sartelet[1]

[1]CEREA, Ecole des Ponts ParisTech, EDF R & D, IPSL, Marne-la-vallée, France
[2]LMD/IPSL, École Polytechnique, Université Paris Saclay, ENS, PSL Research University, Sorbonne Universités, UPMC Univ, France
[3]Université Paris Est Créteil and Univ. Paris Cité, CNRS, LISA, F-94010 Créteil, France
[4]Université Paris Cité and Univ. Paris Est Créteil, CNRS, LISA, F-75013 Paris, France
[5]INERIS, 60550 Verneuil en Halatte, France
[6]Laboratoire des Sciences du Climat et l'Environnement, CEA/Orme des Merisiers, 91191 Gif-sur-Yvette, France
[7]Airparif, 75004, Paris, France
[8]Aix Marseille Univ, CNRS, LCE, Marseille, France

**Correspondence:** Karine Sartelet (karine.sartelet@enpc.fr)

**Abstract.**

This study focuses on mapping the concentrations of pollutants of health interest (NO$_2$, black carbon (BC), PM$_{2.5}$, number of particles (PN)) down to the street scale to represent as accurately as possible the population exposure. Simulations are performed over the Greater Paris area with the WRF-CHIMERE/MUNICH/SSH-aerosol chain, using either the top-down inventory EMEP or the bottom-up inventory Airparif with correction of the traffic flow. The concentrations of the pollutants are higher in streets than in the regional-scale urban background, due to the strong influence of road-traffic emissions locally. Model-to-data comparisons were performed at urban background and traffic stations, and evaluated using two performance criteria from the literature. For BC, harmonized equivalent BC (eBC) concentrations were estimated from concomitant measurements of eBC and elemental carbon. Using the bottom-up inventory with corrected road-traffic flow, the strictest criteria are met for NO$_2$, eBC, PM$_{2.5}$, and PN. Using the EMEP top-down inventory, the strictest criteria are also met for NO$_2$, eBC and PM$_{2.5}$, but errors tend to be larger than with the bottom-up inventory for NO$_2$, eBC and PN. Using the top-down inventory, the concentrations tend to be lower along the streets than those simulated using the bottom-up inventory, especially for NO$_2$ concentrations, resulting in less urban heterogeneities. The impact of the size-distribution of non-exhaust emissions was analyzed at both regional and local scales, and it is higher in heavy-traffic streets. To assess exposure, a french database detailing the number of inhabitants in each building was used. The population-weighted concentration (PWC) was calculated by weighting populations by the outdoor concentrations to which they are exposed at the precise location of their home. An exposure scaling factor (ESF) was determined for each pollutant to estimate the ratio needed to correct urban background concentrations in





order to assess exposure. The average ESF in Paris and Paris Ring Road is higher than 1 for NO₂, eBC, PM₂.₅, PN, because the concentrations simulated at the local scale in streets are higher than those modelled at the regional scale. It indicates that the Parisian population exposure is under-estimated using regional-scale concentrations. Although this underestimation is low for PM$_{2.5}$, with an ESF of 1.04, it is very high for NO$_2$ (1.26), eBC (between 1.22 and 1.24), and PN (1.12). This shows that urban heterogeneities are important to be considered in order to represent the population exposure to NO$_2$, eBC, and PN, but less so for PM$_{2.5}$.

Keywords: multi-scale modelling; population exposure; exposure scaling factor; bottom-up emission inventory; particle number concentration

# 1 Introduction

In metropolises, characterised by densely populated and extensively developed areas, air pollution remains a major concern due to the presence of numerous emission sources, such as traffic, energy consumption, solvents, industrial activities. Traffic emissions receive particular attention because of their influence on local concentrations, with an impact of both exhaust and non-exhaust emissions (Fu et al., 2020; Jereb et al., 2021; Holnicki et al., 2021; Sarica et al., 2023). Environmental regulations aim to reduce key air pollutant concentrations, such as NO$_2$, O$_3$, and fine particulate matter (PM$_{2.5}$). Although a large part of the health impacts are attributed to particles (Southerland et al., 2022), all the compounds of particles do not impact health equally (Park et al., 2018; WHO, 2021). In particular, black carbon (BC) and ultrafine particles (diameter lower than 0.1 $\mu$m) are considered "priority" emerging pollutants (WHO, 2021; Goobie et al., 2024), as they may have large health effects (Lequy et al., 2021; Bouma et al., 2023; Kamińska et al., 2023; Li et al., 2023). Whereas fine particles are best characterized by their mass concentrations, ultrafine particles are best characterized by their particle number (PN) concentrations, because their mass is low compared to those of fine particles.

Many studies have been conducted to assess the effect of emissions and policies to improve air quality in cities (Mao et al., 2005; Yuan et al., 2014; Kuklinska et al., 2015; Selmi et al., 2016; Andre et al., 2020; Lugon et al., 2022), focusing mostly on PM and NO$_2$. Parker et al. (2020) reported a traffic decrease of up to 50% during the COVID-19 pandemic, resulting in a decrease in nitrogen oxides (NOx) and PM$_{2.5}$ concentrations. The reduction in NOx due to decreased vehicle use in 2020 exceeds that observed over the past 20 years, emphasizing the significant influence of traffic on NOx concentrations. BC is also strongly affected by traffic emissions, and the fleet renewal over 10 years leads to a larger reduction of BC concentrations than PM (Andre et al., 2020; Lugon et al., 2022). However, the influences of traffic on BC and PN are rarely evaluated, because they are not regulated, not measured routinely in cities and difficult to model.

Air-quality models represent the mass concentrations of elemental carbon (EC). They may differ from BC concentrations (Savadkoohi et al., 2023), which are usually inferred from optical measurements using aethalometers (Drinovec et al., 2015). Equivalent BC (eBC) is then defined as the mass concentration of BC as indirectly determined by light absorption techniques (Savadkoohi et al., 2024). The BC concentrations could be much larger than the EC one, contributing to large model/measurement discrepancies (Lugon et al., 2021b). A harmonization of BC measurements is needed to allow for direct comparisons





between measured eBC and modeled EC concentrations (Savadkoohi et al., 2024). The PN concentrations are also difficult to model because of the lack of emission inventories and the rapid transformations of the ultrafine particles involved (Kukkonen et al., 2016).

Emission inventories are usually built using either top-down or bottom-up approaches (Guevara et al., 2016). Bottom-up approaches use detailed spatial and temporal information for each activity sector, e.g. number of vehicles for traffic emissions, while top-down approaches use information defined at larger scales (regional or national), which are spatialized using specific data, such as population data. Significant discrepancies may exist between emission inventories using these two approaches (Guevara et al., 2016; Lopez-Aparicio et al., 2017), especially for traffic emissions (Lopez-Aparicio et al., 2017) and non-exhaust emissions from tire, brake and road wear, which have large uncertainties (Piscitello et al., 2021; Tomar et al., 2022). Emission inventories for PN only exist for top-down inventories (Kulmala et al., 2011; Zhong et al., 2023). Sartelet et al. (2022) recently provided a methodology to estimate PN emissions from any emission inventories of PM, making it possible to use either bottom-up and top-down emission inventories.

Regional-scale chemistry-transport models have been used to forecast urban background pollutant concentrations in cities and to conduct prospective studies (Binkowski and Roselle, 2003; Andre et al., 2020). However, because these models do not reflect information on buildings and roads in urban areas, whose scales are much smaller than the grid resolution, they give limited information on the spatial and temporal variations of concentrations in cities. Regional-to-local scale coupled model simulations may provide more detailed concentration distributions and capture local phenomena, such as the high concentrations observed in streets (Kwak et al., 2015; Lee and Kwak, 2020; Park et al., 2021; Lugon et al., 2022; Lin et al., 2023; Wang et al., 2023b; Strömberg et al., 2023). To represent an entire city, multi-scale modeling is often used, coupling a chemistry-transport model and a simple representation of local dispersion (Hood et al., 2018; Karl et al., 2019; Ketzel et al., 2021; Kim et al., 2022). However, most studies use relatively simple chemical mechanisms (NOx - Ox - VOCs chemical mechanism), often ignoring particles or only considering $PM_{2.5}$ as a whole. To model PN, the main difficulty lies in the evaluation of atmospheric transformations (Kukkonen et al., 2016; Strömberg et al., 2023), and there is to our knowledge no multi-scale model currently available to represent PN over a whole city from the urban background down to the street scale taking into account aerosol dynamics.

To simulate gas and particle concentrations over cities from the regional down to the local scale, the street-network Model of Urban Network Intersecting Canyons and Highways (MUNICH) (Kim et al., 2018, 2022) has been coupled with SSH-aerosol (Sartelet et al., 2020) for chemistry and aerosol modelling and to the regional-scale model Polair3D (Lugon et al., 2021a, 2022; Sarica et al., 2023). The Polair3D/MUNICH/SSH-aerosol coupled system takes into account secondary particle formation at all scales consistently with the same chemical and aerosol module, also allowing a non-stationary reactive pollutant dispersion at local scales (Lugon et al., 2022). Although the model was extensively compared to observations at the regional and local scales for $NO_2$, $PM_{2.5}$ and $PM_{10}$ (Sartelet et al., 2018; Lugon et al., 2022; Kim et al., 2022; Sarica et al., 2023), urban multi-scale modelling evaluation of BC and PN at both regional and local scales is still missing.

Many studies have estimated population exposure to air pollutants in urban areas based on measurements or modelling (Bravo et al., 2012; Xie et al., 2017; Khreis et al., 2017; Lugon et al., 2022; Wang et al., 2023a; Chakraborty and Aun, 2023).



To estimate the spatial variations of pollution, Land Use Regression (LUR) models, which are based on measurements, are widely used (Kerckhoffs et al., 2017; Lequy et al., 2022). Chemistry-transport models are often used as well, although their spatial resolution is often coarse (larger than a few km$^2$) (Ostro et al., 2015; Adélaïde et al., 2021), leading to simulated fine PM concentrations much lower than those simulated using LUR models (Lequy et al., 2022). Indeed, the population's exposure is

increased when street-scale aerosol concentrations are used instead of regional-scale concentrations for NO$_2$ and PM$_{2.5}$ (Lugon et al., 2022).

This study aims at defining a methodology for simulating multi-pollutant concentrations, including BC and PN, down to the street scale over Paris, and to estimate the influence of spatial heterogeneities on the population exposure. To do so, the street-network model MUNICH/SSH-aerosol is coupled to the regional-scale model CHIMERE/SSH-aerosol (Wang et al.,

2024; Maison et al., 2024), and evaluated for the modelling of NO$_2$, PM$_{2.5}$, eBC and PN concentrations over the Île-de-France region down to streets of Paris for June and July 2022. A bottom-up emission inventory is used, with detailed traffic emissions classified by emission types, and the traffic flow corrected using traffic counted loops. PN emissions are estimated improving on the methodology of Sartelet et al. (2022). The influence of the bottom-up versus top-down emission inventories is compared, as well as the influence of the size-distribution of non-exhaust emissions. Section 2 describes the model and simulation setup,

including the non-traffic (section 2.2.1), traffic emissions (section 2.2.2), and PN emissions (section 2.2.4), as well as the relative contributions of non-traffic, exhaust, non-exhaust, and other traffic emissions (Section 2.2.5). Section 2.2.3 details the setup for a sensitivity simulation related to the size distribution of non-exhaust emissions. The model is evaluated at background and traffic stations in section 3, and the influence of the emission inventory and the size distribution of non-exhaust emissions is assessed. In section 4, using detailed population data at building level, a scaling exposure factor is determined to estimate

outdoor population exposure for the city of Paris using modelled regional-scale concentrations.

## 2 Material and methods

### 2.1 Model description and simulation setup

The CHIMERE chemistry-transport model (v2020r1) (Menut et al., 2021) is applied over Greater Paris (Île-de-France) to compute atmospheric concentrations of gas-phase and aerosol species considering transport (advection and mixing), deposition,

emissions, chemistry and aerosol dynamics. The chemical scheme used is MELCHIOR2 modified to represent the formation of organic condensables as described in SSH-aerosol v1.3 (Sartelet et al., 2020), which is used for aerosol dynamics (coagulation and condensation/evaporation). The particle size distribution is discretized in 10 sections of diameters between 0.01 and 10 $\mu$m (Appendix A). The CHIMERE model is coupled with the meteorological model Weather and Research Forecasting (WRF) (Powers et al., 2017), which was used to compute the meteorological fields needed in the simulation.

For a fine description of the urban background concentrations over Paris, a fine spatial resolution of 1 km$^2$ is used over Greater Paris (IDF1 domain, Figure 1). Boundary conditions are obtained by simulating three nested domains: the intermediate domain covers the north-west of France with a horizontal resolution of 3 km x 3 km (IDF3 domain), and the outermost domain covers France and western Europe with a horizontal resolution of 9 km x 9 km (FRA9 domain, Figure 1). CAMS





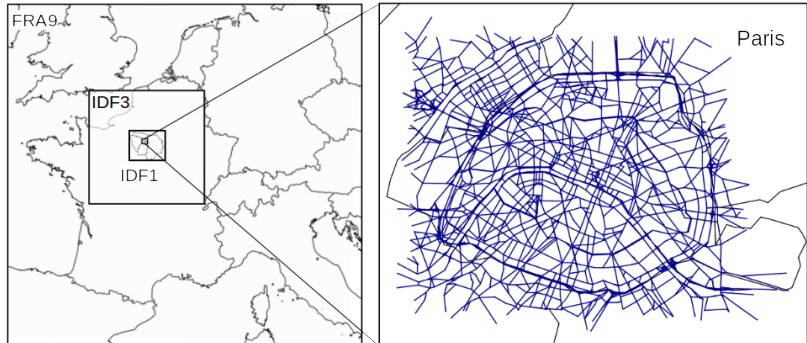

**Figure 1.** Domains for the regional- and local-scale simulations. The blue lines on the right panel represent the street segments used in the local-scale simulation.

reanalysis (Inness et al., 2019) are used as boundary conditions of this outermost domain. The model setup is the same as in
Maison et al. (2024).

To compute meteorological fields, in WRF, the vertical is discretized with 33 levels ranging from 0 to 20 km of altitude. The single-layer urban canopy model (UCM) (Kusaka et al., 2001) is used in the domains IDF3 and IDF1 in order to improve the representation of meteorological fields in urban areas, using the CORINE land cover database (available in 10.2909/71c95a07-e296-44fc-b22b-415f42acfdf0), which classes the urban areas in three sub-categories (commercial areas, high intensity resi-
dential and low-intensity residential areas). More details about WRF simulations setup are available in Maison et al. (2024).

In CHIMERE, the vertical discretization is based on 15 isobaric vertical levels, from 998 hPa to 500 hPa, resulting in average layer heights ranging from 17 m near the ground to 1050 m at high altitudes. The land-use database GLOBCOVER (Arino et al., 2007), with 300 m spatial resolution, is used to compute pollutant deposition and biogenic emissions, which are estimated following MEGANv2.1 (Guenther et al., 2012)

The WRF-CHIMERE/SSH-aerosol model is one-way coupled to the street-network model MUNICH (Kim et al., 2022). SSH-aerosol is used both in CHIMERE and MUNICH, in order to represent consistently chemical and aerosol species at all scales, with the same modified MELCHIOR2 chemistry mechanism. In MUNICH, dry and wet depositions are considered through the parameterizations detailed in Kim et al. (2022). The street network contains the 4655 main streets of Paris (Figure 1). Traffic emissions are diluted in each street volume, which depends on street height, length, and width. Gas and particle
compounds are advected from one street to another, depending on the wind direction, and the vertical transfer of pollutants between the urban background and streets depends on atmospheric stability, which is determined depending on the Monin-Obukhov length (LMO) and Planetary boundary layer (PBL) height, and the standard deviations of wind velocity.

Simulations are performed with WRF-CHIMERE/MUNICH/SSH-aerosol from 01 UTC on 1 June 2022 to 23 UTC on 31 July 2022, corresponding to a period with specific PN and BC measurements performed over Paris, as detailed in section 2.3.
MUNICH v2.2 is employed with an updated calculation of the LMO to better represent the vertical transfer of pollutants between the streets and the background. In previous studies (Kim et al., 2022; Lugon et al., 2022; Sarica et al., 2023), the LMO





was parameterized following Musson-Genon (1992) derived from evaporation and sensible heat fluxes at the surface. Using this parameterization, for all street segments during the period of simulation, 93.8% of atmospheric conditions are unstable, 6.2% are stable, and nearly 0% are neutral. This study uses instead the LMO parameterization of Soulhac et al. (2011), which calculates the sensible heat flux by deriving it from net radiation, latent heat flux, and diffusive heat flux toward the ground. As a result, there are less frequent unstable but more neutral conditions: 68.8% of the conditions are unstable, 7.6% are stable, and 23.6% are neutral. Using both parameterizations, stable and neutral conditions appear predominantly during nighttime. However, using the parameterization of Soulhac et al. (2011) leads to much better comparisons to concentration measurements than the parameterization of Musson-Genon (1992), indicating that stable and neutral conditions may not well be represented during summer in cities with the parameterization of Musson-Genon (1992).

For aerosol dynamics, coagulation and condensation of sulfuric acid and low-volatile organic compounds are resolved simultaneously and dynamically for all particle sizes. The Kelvin effect is taken into account when simulating condensation/evaporation of semi-volatile compounds, as described in Zhu et al. (2016). As condensation/evaporation is resolved using a Lagrangian approach, the section diameters may evolve and the Euler-coupled algorithm is used to redistribute the particle mass and number concentrations into the sections of fixed size classes (Sartelet et al., 2020). Because the first diameter is relatively high (10 nm), if the diameter of the particles of the first section goes below 10 nm, then the mass of those particles is allocated to the first section.

## 2.2  Anthropogenic emissions

In the regional-scale modelling with CHIMERE, the top-down 2020 anthropogenic emission inventory of EMEP (European Monitoring and Evaluation Program) of spatial resolution 0.1° x 0.1° is used for the larger-scale simulations over Western Europe (FRA9 domain) and the North-west of France (IDF3 domain). Over Greater Paris, two different simulations are performed: one with the EMEP inventory, labelled as "EMEP" in the following, and one with the bottom-up inventory of Airparif, the air-quality monitoring network for the Greater Paris area, labelled as "REF" in the following. The spatial resolution of the Airparif inventory is 1 km x 1 km, except for traffic emissions with a finer resolution down to the street level. Note that in the REF simulation, even though the Airparif bottom-up inventory is used over Greater Paris, the EMEP inventory is used for the cells outside the Greater Paris area that are not covered by the Airparif inventory, i.e. exclusively on the edge of the Greater Paris area. In the street-network model MUNICH, only the Airparif inventory is used, because there is no downscaling available for EMEP emissions down to streets.

The bottom-up inventory corresponds to the 2019 Airparif inventory for all activity sectors except for traffic, which is specific to the summer 2022. It is calculated using the traffic emissions model HEAVEN (https://www.airparif.asso.fr/heaven-emissions-du-trafic-en-temps-reel). The strength of this system is to use a traffic model that is corrected from traffic count data in near real time. Non-traffic and traffic emissions are detailed in sections 2.2.1 and 2.2.2, respectively. For consistency between traffic emissions in the streets and at the urban scale, street-scale traffic emissions are aggregated and assigned to the corresponding grids in the CHIMERE model (Figure 4 a and c). Section 2.2.3 details the size distribution of non-exhaust



emissions, and the sensitivity simulation on its representation (labelled as "SEN"). The algorithm to represent PN emissions is detailed in section 2.2.4.

### 2.2.1 Non-Traffic Emissions

Anthropogenic emissions are provided for ten different Standard Nomenclature for Air Pollution (SNAP) categories (Tagaris et al., 2015) for NOx, VOC, CO, $SO_2$, $CH_4$, $PM_{2.5}$, and $PM_{10}$. NOx is speciated as 90% NO and 10% $NO_2$ for all non-traffic

categories (Sartelet et al., 2012; Menut et al., 2021; Lugon et al., 2021a). The speciation of coarse and fine PM emissions follows the CAMS set-up (Kuenen et al., 2021), and volatile organic compounds (VOCs) are speciated following Passant (2002). The emissions of condensables, i.e. the sum of intermediate, semi and low volatility organic compounds (IVOC, SVOC, and LVOC, respectively) are estimated by multiplying primary organic matter emitted in each SNAP by 2.5 (Couvidat et al., 2012; Sartelet et al., 2018). Condensable emissions are then divided into volatility classes (32% in IVOC, 43% in SVOC and 25% in LVOC).

The vertical distribution of non-traffic emissions follows Bieser et al. (2011). In the Airparif inventory, a specific analysis was applied to determine the vertical profiles of plane emissions of NOx, PM, CO, $SO_2$, and VOCs for the three airports (Charles-de-Gaulle (CDG), Paris-Orly (ORY), and Paris–Le Bourget (LBG)) located in Île-de-France (Figure 2). The dataset used for this analysis was obtained from detailed information on flight trajectories for each airport. Specifically, the flight paths were represented as clouds of points in three dimensions (x, y, z), where the altitude (z) was rounded to the nearest 25 meters.

The emissions from each airport and pollutant were then summed up to determine the altitudes at which they were produced, and this information was used to create vertical profiles of emissions. These vertical profiles are used to distribute vertically the emissions on the CHIMERE vertical levels. Figure 2 shows the vertical $NO_2$ and BC emissions profiles at the three airports. Note that the average $NO_2$ (0.37 $\mu$g m$^{-2}$ s$^{-1}$) and BC (0.04 $\mu$g m$^{-2}$ s$^{-1}$) emissions at the lowest height near the airports are much lower than vehicle emissions of $NO_2$ (1.65 $\mu$g m$^{-2}$ s$^{-1}$) and BC (0.08 $\mu$g m$^{-2}$ s$^{-1}$) averaged over Greater Paris.

### 2.2.2 Traffic emissions

The road traffic emissions data were calculated by Airparif using the HEAVEN system, originally developed in 2001 as part of the European project of the same name in partnership with the road traffic management departments of the City of Paris and the Direction Régionale de l'Equipement d'Ile-de-France. Since then, this system has been regularly updated on all its components: emission factors, vehicle fleets, traffic model, real-time counting, network, etc., in order to have the most recent

information on vehicle emissions in the Paris region.

The traffic emissions are calculated hourly for each street and road segment for June and July 2022 over Île-de-France. The traffic emissions inventory was categorized based on emission types (combustion, tire wear, road wear, brake wear, and evaporation), fuel types (electric, gasoline, diesel, Liquefied Petroleum Gas, and Compressed Natural Gas), vehicle types, and Crit'Air classification. In France, the Crit'Air sticker classifies vehicles according to the fine particles and levels of nitrogen ox-

ide that they emit. The vehicle types include passenger vehicles (VP), utility vehicles (VU), heavy vehicles (PL), trailers (TC), and two-wheeled vehicles (2R). The Crit'Air classification depends on the fuel type and the age of the vehicle, as determined

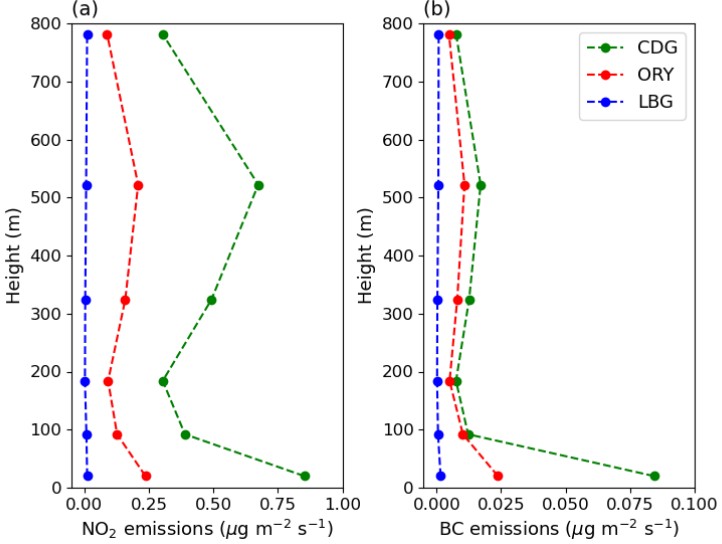

**Figure 2.** Vertical profiles of the average (a) NO$_2$ and (b) BC emissions at the airports.

from the Euro Norm. The European emission standards (Euro Norm) of the vehicles are estimated from the information on fuel type and Crit'Air classification. Information about the different categories is used to speciate the emissions of the inventory.

The NOx emissions on each road are speciated into NO, NO$_2$, and HONO, depending on the Euro Norm of the vehicles and
using speciation from the EMEP guidelines (Leonidas Ntziachristos, 2023b). The average speciation into NO, NO$_2$, and HONO for traffic exhaust emission is 88.5%, 10.6%, and 0.9%, respectively. The traffic emissions of NO$_2$ and BC are illustrated in Figure 3. As expected, emissions are higher during rush hours. Also, NO$_2$ and BC emissions at the BP_EST station, which is next to a heavy-traffic road, are higher than at the HAUSS station (City center). At the city center, NO$_2$ emissions are clearly higher during weekdays than on weekends.

The traffic PM emitted from combustion are assumed to be of diameters lower than 1 $\mu$m (PM$_1$) and to be composed of organic matter (OM) and black carbon (BC). The OM/BC ratios differ according to the vehicle fuel, and follow the values proposed in Kostenidou et al. (2021): PM$_{2.5}$ emissions are speciated as 75% of BC and 25% of OM for diesel, while 25% of BC and 75% of OM for gasoline and LPG. The traffic PM emitted from non-exhaust sources are speciated in OM, BC, and inert particles as described in the EMEP/EEA air pollutant emission inventory guidebook. The speciation of Non-Methane
VOC follows Theloke and Friedrich (2007), which defines different speciations for gasoline and diesel-powered vehicles. For traffic emissions, the emitted OM is assumed to correspond to LVOC. IVOC and SVOC speciated are speciated based on ratios of Non-methane hydrocarbon emissions, which varies according to the vehicle fuel (Sarica et al., 2023).



The PM emitted from non-exhaust sources (tire, brake, and road wear) are classified as fine ($PM_1$, $PM_{1-2.5}$) and coarse ($PM_{2.5-10}$) particles. They are speciated according to the EMEP guideline (Leonidas Ntziachristos, 2023a), as detailed in Table 1.

Specifically for the simulations using the EMEP emission inventory, as no detailed information is available regarding exhaust and non-exhaust emissions, as well as vehicle categories, the CAMS traffic speciation is employed for PM, and NOx emissions are assumed 90% of NO and 10% of $NO_2$. Also, the total IVOC, SVOC, and LVOC are estimated by multiplying primary organic matter emitted from traffic by 2.5 (Couvidat et al., 2012; Sartelet et al., 2018). For traffic emissions, this total is splitted between 51% of IVOC, 14% of SVOC, and 35% of LVOC.

**Table 1.** Chemical composition of particles emitted from traffic non-exhaust sources (Leonidas Ntziachristos, 2023a)

|  | $PM_1$ | | | $PM_{1-2.5}$ | | | $PM_{2.5-10}$ | | |
|---|---|---|---|---|---|---|---|---|---|
|  | Tire | Brake | Road | Tire | Brake | Road | Tire | Brake | Road |
| BC (%) | 30 | 80 | 0 | 21 | 2 | 0 | 0 | 0 | 0 |
| OM (%) | 70 | 20 | 0 | 48 | 8 | 0 | 0 | 0 | 0 |
| Dust (%) | 0 | 0 | 100 | 31 | 90 | 100 | 100 | 100 | 100 |

Figure 4 shows maps of $NO_2$ and BC traffic emissions over Greater Paris with a zoom over Paris. The emissions are higher along the main roads and motorways. In Paris, emissions are higher on the Paris Ring Road than in the city center. Although street-scale emissions are directly used as input of the street-network model MUNICH, emissions over Greater Paris are gridded, such as being used as input of the CHIMERE model to simulate urban background concentrations.

### 2.2.3 Sensitivity to non-exhaust emissions size distribution

Non-exhaust traffic emissions, such as tire wear, brake, and road wear, have large uncertainties (Lugon et al., 2021b), their contribution to particle emissions is also large. Section 2.2.5 shows that non-exhaust emissions (22.2%) contribute more to traffic PM emissions than exhaust emissions (8.3%). A sensitivity simulation "SEN" is set-up to evaluate the impact of choices made in the size distribution of non-exhaust emissions. In the reference simulation, the $PM_1/PM_{2.5}$ ratio and $PM_{2.5}/PM_{10}$ ratio for non-exhaust emissions are obtained from EMEP guidelines for the different types of vehicles (Ntziachristos, 2016). In the SEN scenario, the $PM_{2.5}/PM_{10}$ ratios are modified to be those of the NORTRIP model (Denby et al., 2013). The size distribution employed in REF and SEN simulations is detailed in Table 2.

**Table 2.** $PM_{2.5}/PM_{10}$ ratios of particles emitted from non-exhaust traffic sources in the REF and SEN simulations.

|  | Tire | Brake | Road |
|---|---|---|---|
| REF | 0.70 | 0.40 | 0.54 |
| SEN | 0.10 | 0.63 | 0.04 |





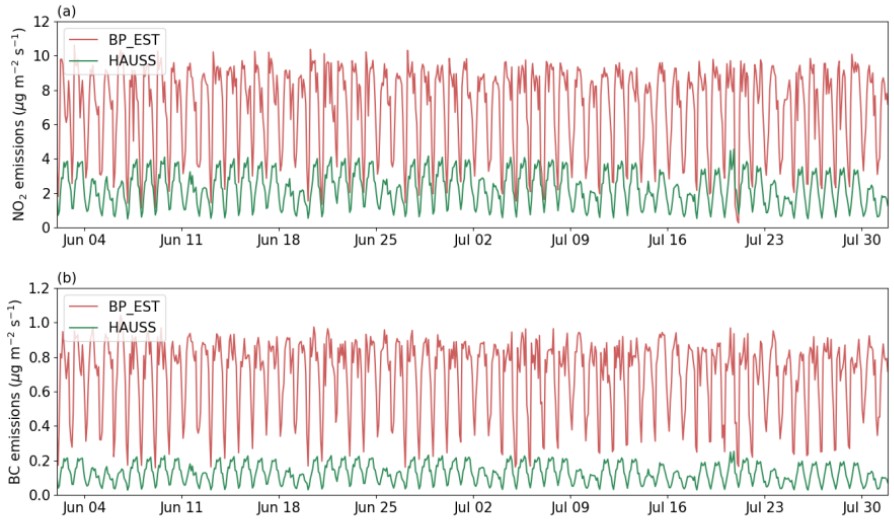

**Figure 3.** Time series for (a) NO$_2$ and (b) BC emissions at the HAUSS (city center) and BP_EST (heavy-traffic) stations

Non-exhaust emissions are almost equally distributed between PM$_{2.5}$ and PM$_{10}$ in the REF simulation (PM$_{2.5}$/PM$_{10}$ = 0.53), while the proportion of non-exhaust emissions in PM$_{2.5}$ is lower in the SEN simulation (PM$_{2.5}$/PM$_{10}$ = 0.28). As the NORTRIP model has no PM$_1$/PM$_{2.5}$ ratio for non-exhaust emissions, no modifications are made in the PM$_1$/PM$_{2.5}$ ratios in the SEN simulation. The proportion of PM$_1$ is low in both REF and SEN (PM$_1$/PM$_{10}$ $\simeq$ 0.07).

### 2.2.4 PN emissions

The emission inventories provide estimations of PM$_{2.5}$ and PM$_{10}$ emissions for the different activity sectors. The emissions of PM$_{2.5}$ are distributed in the modelled particle size sections following Sartelet et al. (2022). First, emissions of particles in the range PM$_{0.1}$-PM$_1$ and PM$_{0.01}$-PM$_{0.1}$ are estimated using the PM$_1$/PM$_{2.5}$ and PM$_{0.1}$/PM$_1$ ratios given in Sartelet et al. (2022) (Table A1) for each activity sector.

Emissions in each of the size ranges: PM$_{0.01}$-PM$_{0.1}$, PM$_{0.1}$-PM$_1$, and PM$_1$-PM$_{2.5}$ are distributed amongst the model size sections within that range using the formula:

$$M_i = M \frac{d_i^{3/2} - d_{i-1}^{3/2}}{d_n^{3/2} - d_0^{3/2}}$$

where $M$ is the mass concentration to be distributed, $i$ is the size section of bound diameters $d_{i-1}$-$d_i$ included in the size range of bound diameters $d_0$-$d_n$. This formula, which is set out in Appendix A, allows the conservation of both mass and number during the discretisation. The detailed factors obtained with this formula for each size section are presented in Table A1.





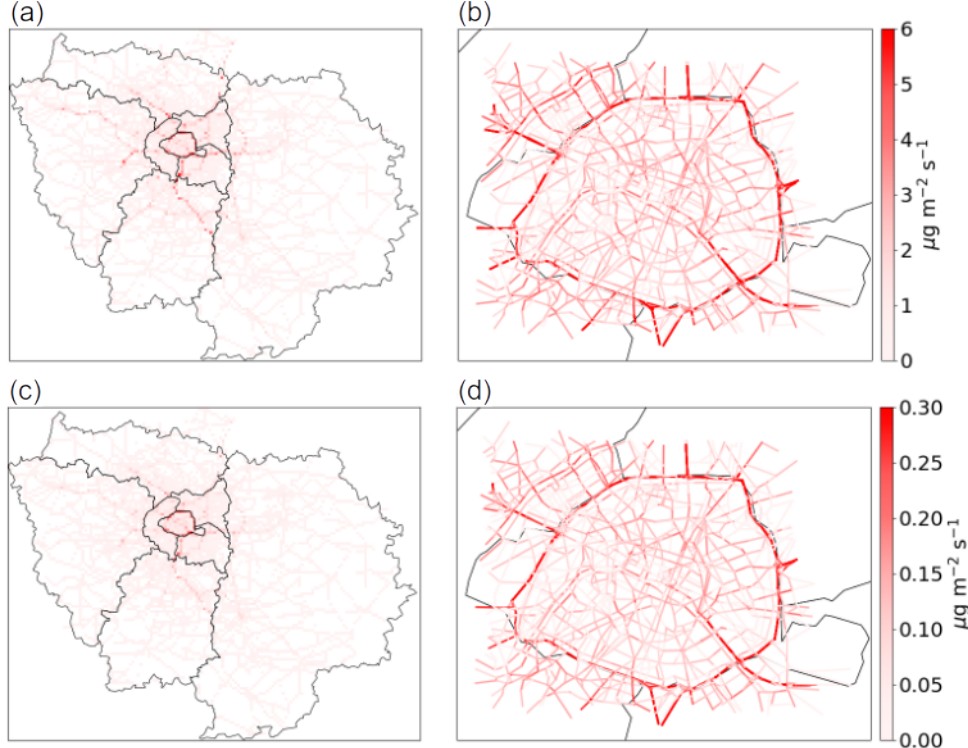

**Figure 4.** Bottom-up (left panels) and street (right panels) traffic emissions of NO₂ [(a) and (b)] and BC [(c) and (d)] averaged from June to July 2022

### 2.2.5 Emission sources of BC, PM, and PN

The distribution between traffic and non-traffic emissions for NOx, BC, PM, and PN over Greater Paris is shown in Figure 5,
and over Paris in Figure B3. Over Greater Paris, the majority of NOx emissions originated from road traffic sources (65.8%).
Amongst non-road traffic sources, shipping, railways, and aviation contribute to a significant proportion (24.9%) of NOx
emissions. PM$_{2.5}$ emissions are mostly from non-traffic sources (over 69% of PM emissions), while road traffic contributes only
to about 30.5% of emissions. Note that non-exhaust emissions represent a larger percentage of PM$_{2.5}$ than exhaust emissions
(22.2% vs 8.3%). Concerning BC emissions, the influence of road traffic is larger than that of PM$_{2.5}$ emissions (about 46.1% of
265 BC emissions), with a larger contribution of exhaust than non-exhaust emissions (32.5% vs 13.6%). Concerning PN emissions,
non-traffic sources constitute almost half of the emissions (52.3%), with a contribution of road traffic to about 17.2%, and a
strong contribution of non-road traffic sources (24.5%).



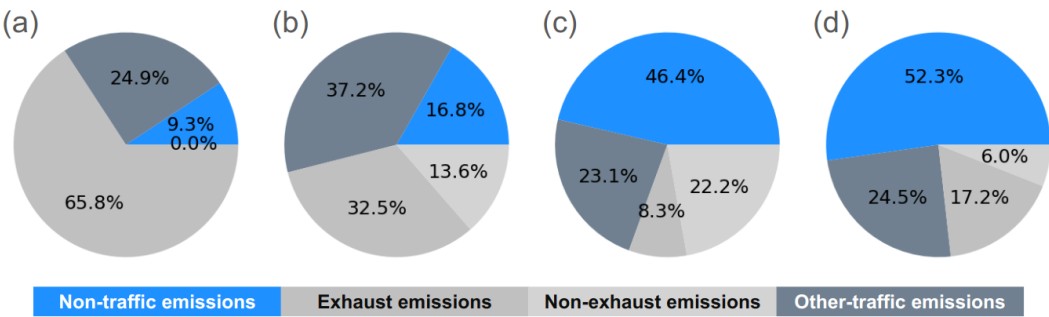

**Figure 5.** Distribution between vehicular traffic (exhaust and non-exhaust), other traffic and non-traffic emissions for NOx (a), BC (b), PM$_{2.5}$ (c), and PN (d) in Greater Paris.

## 2.3 Measurements

The simulated concentrations are compared to measurements from different categories of monitoring stations: rural, suburban, urban background, and traffic (Figure B1). Concerning the measurements operated by Airparif, NO$_2$ and PM$_{2.5}$ are routinely monitored at 21 and 8 stations, respectively. BC was monitored at 4 stations using a dual spot aethalometer (MAGE Sci. model AE33 M8060, using the optical wavelengths 880 nm for eBC analysis) with a PM$_{2.5}$ size cut off, and PN was monitored using a Mobility Particle Size Spectrometer (MPSS) for particles of diameters between 5 nm and 385 nm at two stations: Châtelet-les-Halles (PA01H), which is an urban background station located in the center of Paris, and BP-EST, which is next to the very busy Ring road of Paris. At the PA01H station, EC was also measured using filter measurements. It is analysed according to a thermo-optical method (Instrument : Laboratory OCEC Carbon Aerosol Analyser, SUNSET), which meets the EUSAAR2 protocol. The concomittant EC and eBC measurements allow the determination of the harmonization factor suggested by Savadkoohi et al. (2024).

The harmonization factor was calculated as the ratio of eBC and EC. Between 9 and 22 June 2022, at the PA01H station where both EC and eBC data are available, the daily harmonization factor ranges from 1.34 to 2.26 with an average equal to 1.79 (Figure B2). This average harmonization factor is used to correct the eBC measured concentrations at all stations and all times. Note that the average harmonization factor (1.79) is similar to the harmonization factor of 1.76 proposed by Savadkoohi et al. (2023).

Comparisons are also performed at the SIRTA site (Site Instrumental de Recherche par Télédétection Atmosphérique), an atmospheric observatory located 20 km south-west of Paris which is integrated in the ACTRIS European Research Infrastructure Consortium (https://www.actris.eu) (Haeffelin et al., 2005). In summer 2022, two other monitoring stations were operated inside Paris: the Paris Rive Gauche (PRG) urban background station and the Hôtel de Ville (HdV) traffic station. The PRG site, located on the 7[th] floor of the Lamark B building of Université Paris Cité (30 m above ground layer), in the south-east side of the city, was set up as part of the ACROSS (Atmospheric ChemistRy Of the Suburban foreSt) campaign (Cantrell and Michoud, 2022). Air was taken from a common certified PM$_1$ sampling head located on the roof of the building at about 30 m



high above the ground level. A copper tube of 7 mm inner diameter and 10 m length connected the $PM_1$ head to a 4–way stainless steel flow splitter (TSI) located in the measuring room that dispatched the airflow to different instruments. The Paris HdV station was setup as part of the sTREEt (Impact of sTress on uRban trEEs and on city air quality) project (Maison et al., 2024). It overlooks the quays of the Seine, which is a major road traffic artery. The eBC was monitored with an AE33 at all
295 stations (MAGE Sci. model AE33 M8060, using the optical wavelengths 880 nm), with a $PM_{2.5}$ size cut off at SIRTA and HdV, and a with a $PM_1$ size cut off at PRG. The PN was monitored with an SMPS for particles of diameters between 9 nm and 836 nm at SIRTA, and between 23 nm and 1 $\mu$m at PRG. A "nano" SMPS, measuring in the range 8 nm - 64 nm was used at PRG to supplement measurements for diameters between 8 and 23 nm. Note that for comparison between simulated and measured concentrations, only particles of diameters larger than 10 nm are considered. The PN measurements at PRG and
300 SIRTA were performed for shorter periods than the Aiparif measurements at PA01: from 17 June to 11 July at PRG, and from 15 June to 17 July at SIRTA.

## 3 Evaluation of model performances

Model to data comparisons are performed over urban background stations to evaluate the regional-scale concentrations simulated with CHIMERE, and over traffic stations to evaluate the street-scale concentrations simulated with MUNICH. Model
performances are evaluated based on statistical indicators, including the fractional bias (FB), geometric mean bias (MG), normalised mean square error (NMSE), geometric variance (VG), normalised absolute difference (NAD), and the fraction of predictions within a factor of 2 of observations (FAC2), as described in Section C. Following Hanna and Chang (2012) and Herring and Huq (2018), two different acceptable criteria are considered: ($i$) a strict performance criteria, with |FB| < 0.3, 0.7 < MG < 1.3, NMSE < 3, VG < 1.6, NAD < 0.3, and FAC2 > 0.5; and ($ii$) a less strict performance criteria, acceptable for urban
areas, with |FB| < 0.67, NMSE < 6, NAD < 0.5, and FAC2 > 0.3. For evaluating PN concentrations, many studies (Olin et al., 2022; Patoulias and Pandis, 2022a; Sartelet et al., 2022) rather used the NMB, which is the normalized mean bias, and NME the normalized mean error.

### 3.1 Urban background concentrations

The average statistical indicators obtained in each CHIMERE simulation (REF, SEN, and EMEP simulations) are presented
in Table 3. The most strict criteria are also met for all pollutants in the simulations using the bottom-up emission inventory (REF and SEN simulations). In the EMEP simulation, concentrations tend to be over-estimated for all pollutants, and the $NO_2$, eBC, $PM_{2.5}$, and PN concentrations are higher than those in the REF and SEN simulations (Figure B5) due to relatively high emissions compared to bottom-up traffic emissions. For $NO_2$, eBC, $PM_{2.5}$, and PN, the emission ratios of EMEP to bottom-up emissions are 1.2, 1.72, 1.12, and 1.53 in Greater Paris, respectively.
As the sensitivity simulation differs from the reference one by the size distribution of non-exhaust emissions between $PM_{2.5}$ and $PM_{10}$, the differences between the REF and SEN simulations are negligible for all pollutants. They are also very low for $PM_{2.5}$, suggesting that the choice of the EMEP or NORTRIP size distribution of non-exhaust traffic emissions has a low



influence at the regional scale. The differences are the highest for eBC, as the speciation of non-exhaust emissions differs for $PM_{2.5}$ and $PM_{10}$ (see Table 1).

**Table 3.** Statistical indicators for $NO_2$, eBC, $PM_{2.5}$, and PN concentrations simulated at the background stations in the REF, SEN, and EMEP simulations. The average simulated and observed concentrations are in $\mu g\ m^{-3}$ for $NO_2$, eBC, $PM_{2.5}$, and in $\#\ cm^{-3}$ for PN. Bold values do not respect more strict performance criteria.

| | NO_2 | | eBC | | | PM_2.5 | | | PN | | |
|---|---|---|---|---|---|---|---|---|---|---|---|
| | REF/SEN | EMEP | REF | SEN | EMEP | REF | SEN | EMEP | REF | SEN | EMEP |
| Number of stations | 21 | | 4 | | | 8 | | | 3 | | |
| Observation | 14.97 | | 0.44 | | | 7.20 | | | 8176 | | |
| Simulation | 14.74 | 17.50 | 0.36 | 0.35 | 0.54 | 6.76 | 6.67 | 7.29 | 8173 | 8138 | 10610 |
| FB | -0.03 | 0.15 | -0.15 | -0.19 | 0.24 | -0.05 | -0.06 | 0.03 | -0.01 | -0.01 | 0.23 |
| MG | 1.04 | 1.24 | 0.97 | 0.94 | 0.82 | 1.04 | 1.02 | 1.11 | 1.05 | 1.05 | **1.34** |
| NMSE | 0.21 | 0.20 | 0.35 | 0.38 | 0.28 | 0.20 | 0.20 | 0.18 | 0.10 | 0.10 | 0.16 |
| VG | 1.21 | 1.24 | 1.30 | 1.32 | 0.80 | 1.17 | 1.18 | 1.18 | 1.18 | 1.18 | 1.28 |
| NAD | 0.17 | 0.17 | 0.21 | 0.22 | 0.20 | 0.16 | 0.16 | 0.15 | 0.12 | 0.12 | 0.16 |
| FAC2 | 0.90 | 0.87 | 0.80 | 0.80 | 0.79 | 0.92 | 0.92 | 0.91 | 0.98 | 0.98 | 0.89 |
| NMB (%) | 0.1 | 19 | -10 | -13 | 34 | -2 | -3 | 6 | -0 | -1 | 27 |
| NME (%) | 33.5 | 38 | 39 | 40 | 50 | 31 | 31 | 32 | 24 | 24 | 37 |

The simulation performance regarding PN concentrations is also evaluated based on NMB and NME, following Olin et al. (2022), Patoulias and Pandis (2022a), and Sartelet et al. (2022). The concentrations simulated with the bottom-up inventory presented very low bias NMB (between -0.3% for REF and -0.7% for SEN) and low NME (24%). Using the EMEP top-down inventory, the NMB and NME are higher (27% and 37% respectively). These values are low compared to studies in the literature. For example, for particles of diameters larger than 0.01 $\mu$m, NME values range between 36% and 79% (Sartelet et al., 2022), 63% (Patoulias and Pandis, 2022b), 94% (Olin et al., 2022), and Ketzel et al. (2021) reported an NMB of 151%.

Figure 6 shows the model-to-data comparison of PN concentration for different size sections. All simulations reproduced well the size distribution of PN concentrations, although the concentrations are overestimated in the first size section, between 10 and 20 nm. This overestimation may be partly due to the redistribution algorithm, where the mass of particles whose diameter becomes lower than 10 nm during the simulation is assigned to the first size section, as detailed in section 2.1.

The better performance of REF and SEN simulations compared to EMEP is more visible in dense-urban areas, probably because the share of traffic is more important than in less dense areas and traffic emissions are better represented thanks to the correction of traffic emissions based on counting loops (see Section 2.1). Figure 7 shows the daily time series for the $NO_2$, eBC, $PM_{2.5}$, and PN in the city center (PA01H station) measured and simulated with the three CHIMERE simulations. Especially for eBC, the temporal evolution of pollutants is better represented with the REF and SEN simulations than the EMEP one, enhancing the importance of a precise traffic emission inventory. Large concentration peaks of $PM_{2.5}$ and eBC are




observed at all stations on 19 July. They could explained by forest fires in the South West of France (Menut et al., 2023). Apart from this peak, in both the REF and SEN simulations, the daily variations of $NO_2$, eBC, $PM_{2.5}$, and PN concentrations are well simulated. As observed for the statistical indicators, the effect of changing the size distribution of non-exhaust emissions (SEN simulation) is low on eBC, $PM_{2.5}$, and PN concentrations at urban background stations, and using the EMEP emission inventory leads to an overestimation of concentrations.

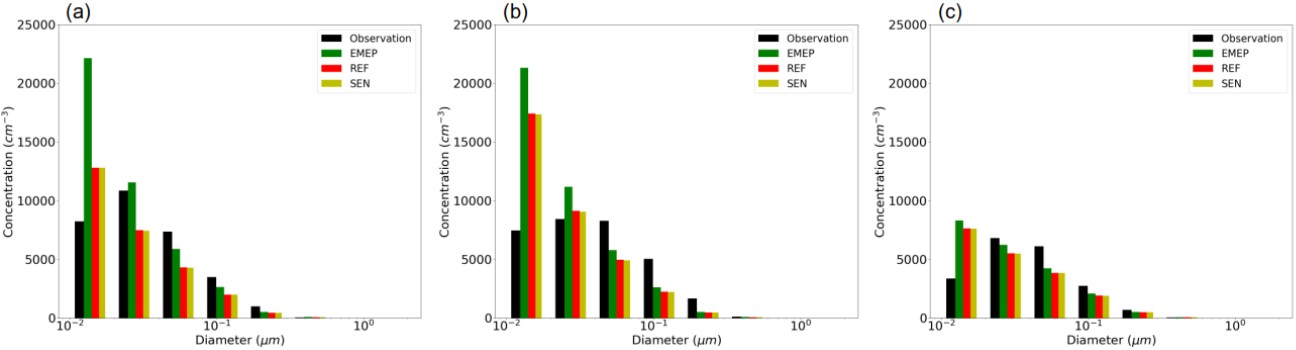

**Figure 6.** Size distribution of PN concentrations at (a) PA01H, (b) PRG, and (c) SIRTA stations

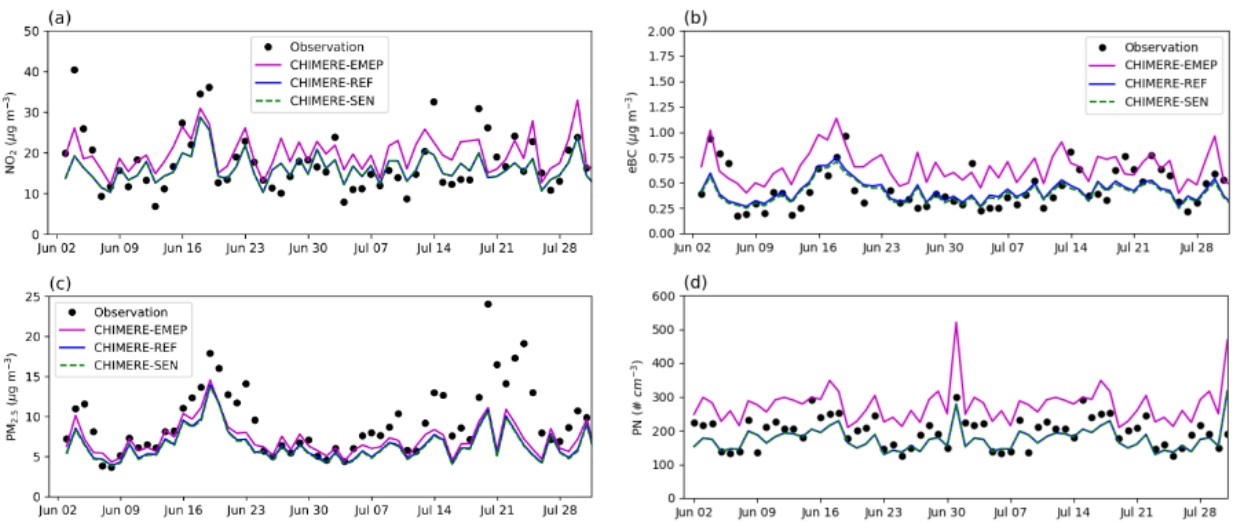

**Figure 7.** Time series for measured and simulated (a) $NO_2$, (b) eBC, (c) $PM_{2.5}$, and (d) PN concentrations at PA01H station





## 3.2 Street concentrations

Table 4 summarizes the statistical indicators for $NO_2$, eBC, and $PM_{2.5}$ concentrations at the traffic stations. The strict performance criteria are met for all pollutants: $NO_2$, eBC, and $PM_{2.5}$ concentrations, for both the REF and SEN simulations.

Figure 8 shows the daily temporal variations of the concentrations of $NO_2$, eBC, and $PM_{2.5}$ at two different traffic stations:
HAUSS, representing a street in the city center, and BP_EST, representing a heavy-traffic road. As expected, the concentrations
are largely underestimated by the regional-scale concentrations simulated with CHIMERE. The daily variations of the street
concentrations simulated by MUNICH are strongly influenced by those of CHIMERE. For example, the peak of concentrations
around 19 July is also present at the street scale, but underestimated by the model. In the SEN simulations, eBC and $PM_{2.5}$
concentrations are lower than in the REF simulation due to the reduction of the non-exhaust emissions of small-size sections
(0.01 to 2.5 $\mu$m). The largest difference are observed at the heavy-traffic station BP_EST (Figure 8 f ). Large peaks of eBC
concentrations are observed at the BP_EST station. There is greater variability in the measured eBC concentrations compared
to those of $NO_2$ at the BP_EST station. This could be due to strong variations in the traffic BC emission factors, or to temporal
variations in the EC/BC ratio, which is assumed to be constant.

For PN, due to the lack of PN measurement at traffic stations in 2022, for indicative purposes, the concentrations simulated
in summer 2022 are compared with the concentrations observed in summer 2021 at two traffic sites (BP_EST and HAUSS),
as documented in the Airparif report (AIRPARIF, 2022). The measurement campaign operated by Airparif took place from
14 June to 19 September 2021 (AIRPARIF, 2022). At PA01H station, the average PN concentration for particles of diameters
between 10 and 400 nm is very close to the average measured in 2022 (about 9,300 # $cm^{-3}$ both in 2021 and 2022). At
the two traffic stations (HAUSS and BP_EST), the average PN concentrations measured in the summer 2021 were 17,000
# $cm^{-3}$ and 21,300 # $cm^{-3}$ respectively for particles of diameters between 10 and 400 nm. The PN concentrations in the REF
(SEN) simulation are 14,500 (14,400) # $cm^{-3}$ at HAUSS and 30,650 (29,800) $cm^{-3}$ at BP_EST, suggesting a bias of 15% at
HAUSS and 40% at BP_EST. The wider bias at BP_EST is certainly due to the 10 nm cut-off diameter for the particles. At
HAUSS, most of the particle number concentrations is between 10 nm and 400 nm, which represent 94% of particles between
5 and 400 nm. However, at BP_EST, many particles are observed between 5 and 10 nm because of the higher importance of
vehicle emissions. The number concentration between 10 and 400 nm represent 86% of the particles between 5 and 400 nm.
Although only particles with diameters greater than 10 nm are modelled here, a proportion of particles with smaller diameters
is represented, since particles with diameters less than 10 nm are assigned to the first size section (10-20 nm) in the numerical
algorithm used here. Although there is a difference in the periods of measurement and simulation with general conditions
(traffic emissions, meteorological parameters) that could have been different, the simulated PN concentrations are roughly
consistent with the measured ones, and the concentrations in the streets and at the traffic sites are much higher than those in
the urban background.



**Table 4.** Statistical indicators for $NO_2$, eBC, and $PM_{2.5}$ concentrations simulated at the traffic stations in the REF and SEN simulations. The average simulated and observed concentrations are in $\mu g\ m^{-3}$ for $NO_2$, eBC, and $PM_{2.5}$. All values respect the strict performance criteria.

|  | $NO_2$ | eBC | | $PM_{2.5}$ | |
| --- | --- | --- | --- | --- | --- |
|  | REF/SEN | REF | SEN | REF | SEN |
| Number of stations | 10 | 3 | | 3 | |
| Observation | 40.04 | 1.26 | | 11.21 | |
| Simulation | 38.33 | 1.26 | 1.09 | 9.91 | 8.56 |
| FB | -0.02 | 0.09 | -0.05 | -0.13 | -0.27 |
| MG | 1.00 | 0.87 | 1.00 | 1.14 | 1.30 |
| NMSE | 0.16 | 0.20 | 0.23 | 0.12 | 0.17 |
| VG | 1.17 | 1.23 | 1.21 | 1.12 | 1.16 |
| NAD | 0.15 | 0.19 | 0.19 | 0.14 | 0.16 |
| FAC2 | 0.92 | 0.89 | 0.90 | 0.95 | 0.95 |
| NMB (%) | 1 | 11 | -2 | -10 | -22 |
| NME (%) | 30 | 40 | 37 | 26 | 29 |

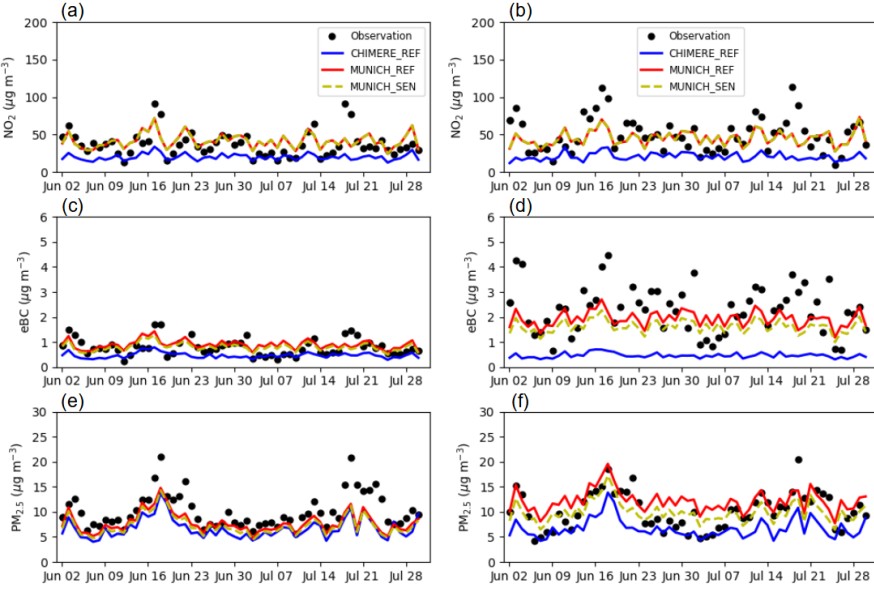

**Figure 8.** Time series of measured and simulated $NO_2$ [(a) and (b)], eBC [(c) and (d)], and $PM_{2.5}$ [(e) and (f)] concentrations at the HAUSS (left panels) and BP_EST (right panels) stations.





### 3.3 Impact of the emission inventory over Greater Paris

The impacts of the emission inventory are investigated over Greater Paris. In the EMEP simulation, the $NO_2$ concentrations are lower along the roads and airport than those in the REF simulation due to higher traffic emissions using the bottom-up inventory than the EMEP one (Figure B5). In contrast, $NO_2$ concentrations in areas excluding roads are higher in the EMEP simulation. The differences in the spatial distributions of eBC and $PM_{2.5}$ concentrations are similar to those of $NO_2$ concentrations, but the spatial differences are less pronounced than for $NO_2$. In the eastern region of Greater Paris and the extreme northwest part of the region, PN concentrations are lower using EMEP than the bottom-up inventory owing to lower emissions compared to other areas within the region.

Figure 9 shows the average $NO_2$, eBC, $PM_{2.5}$, and PN concentrations in the REF simulation over Greater Paris with a zoom over Paris down to the street scale. The concentrations are the highest over the center of Paris, and in the main streets of Paris. The spatial gradients of concentrations are stronger for $NO_2$, eBC, and PN than for $PM_{2.5}$. $PM_{2.5}$ concentrations are highest over Paris, but high concentrations are also observed at different locations, such as over large woods, due to the formation of secondary organic aerosols. As the lifetime of $PM_{2.5}$ is about a week (Seigneur, 2019), concentrations are relatively homogeneous and high all over Greater Paris. For $NO_2$, eBC, and PN, the concentrations are much higher in the main streets than in the urban background than in the suburban and rural areas. The concentrations' spatial gradients may strongly influence the estimation of the population exposure, as detailed in the next section.

### 4 Population exposure

The population exposure to outdoor concentrations may be calculated using regional-scale modelling, potentially under-estimating the exposure. To estimate how well the population exposure is represented using regional-scale modelling for the different pollutants, the population-weighted concentration (PWC) and exposure scaling factor (ESF) are calculated for Paris using the multi-scale concentrations and the population data from the MAJIC database (Létinois, 2014). The 'MA-JIC' spatialization method provides a very detailed description of the population on a local scale. More specifically, it uses data on residential premises from the MAJIC property database issued by the French Public Finance Department (DGFiP). This data is cross-referenced with IGN spatial databases (BD PARCELLAIRE https://geoservices.ign.fr/bdparcellaire and BD TOPO, https://geoservices.ign.fr/documentation/donnees/vecteur/bdtopo) and population statistics from the National Institute of Statistics and Economic Studies (INSEE) to estimate the number of inhabitants in each building.

To locate the buildings in the street network, streets are considered as rectangles of known four extremity coordinates, and buildings are located into the street segments by comparing their mean coordinates with those of streets. Because of uncertainty about street widths, buildings within a perimeter on each side of the street are integrated into the street segment (Figure B6).

Amongst the 109,152 buildings of Paris, 66.9% are integrated into street segments. This corresponds to 44.4% of the population living near main streets, and the remaining 55.6% residing away from main streets.




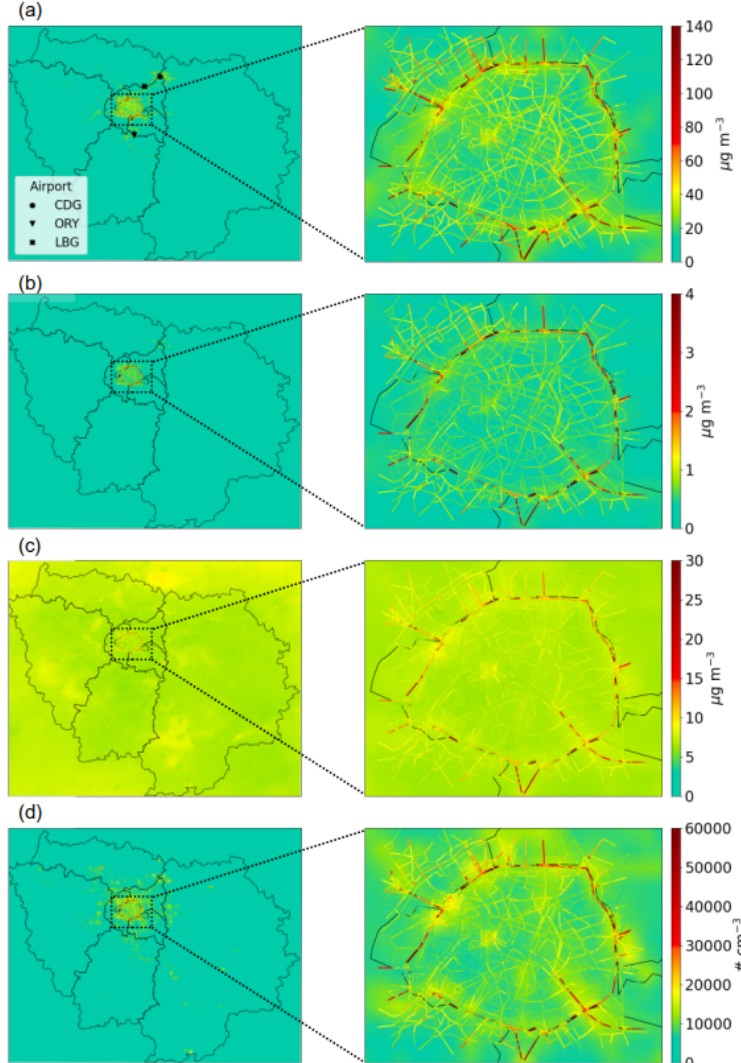

**Figure 9.** Concentration distribution of the average (a) NO$_2$, (b) eBC, (c) PM$_{2.5}$, and (d) PN concentrations over the period in the REF simulation.

PWC is calculated by weighting populations based on the concentrations to which they are exposed at the precise location of their home. People living in a building on the main street are assigned to that street concentration. People living in buildings away from busy streets are assigned to urban background concentrations.

$$PWC_{I,J} = \frac{(POP \cdot C)_{I,J}^{st} + POP_{I,J}^{bg} \cdot C_{I,J}^{bg}}{POP_{I,J}^{st} + POP_{I,J}^{bg}} \tag{1}$$



with

$$(POP \cdot C)^{st}_{I,J} = \sum_{s,building \in I,J} POP^{building,s} \cdot C^{st}(s) \tag{2}$$

where $I$ and $J$ are the grid points of the regional-scale model, $C^{st}$ and $C^{bg}$ are the street and urban background concentra-
tions, respectively. $POP^{building,s}$ is the population living in the street segment. $POP^{st}_{I,J}$ is the population living near main
streets in the grid cell $I$, $J$, $POP^{bg}$ is the population not living near main streets, which is assumed to be exposed to urban
background concentrations.

The spatial distributions of PWC for $NO_2$, eBC, $PM_{2.5}$, and PN are similar to their concentrations (Figure 10), i.e. they have
larger values in grid cells with main traffic roads. The average PWC in Paris for $NO_2$, eBC, $PM_{2.5}$, and PN are 25.3 $\mu$g m$^{-3}$,
0.6 $\mu$g m$^{-3}$, 7.1 $\mu$g m$^{-3}$, and 10963 $cm^{-3}$ in the REF simulation, respectively (Table 5). PWC for eBC, $PM_{2.5}$, and PN in
the SEN simulation is lower than in the REF simulation. The elevated PWC for $NO_2$, eBC, $PM_{2.5}$, and PN are observed in
heavy-traffic regions in both simulations.

ESF is defined as the ratio of PWC to the regional-scale concentrations simulated by CHIMERE. ESF is higher than 1 for
all pollutants in most areas of Paris (Figure 11). The average ESF over Paris is the highest for $NO_2$, followed by eBC, PN, and
finally $PM_{2.5}$ (Table 5). For $PM_{2.5}$, the ESF is close to 1 in Paris (1.04 in the REF simulation and 1.02 in the SEN simulation),
because of the low differences between regional and local scale concentrations (Figure 9 c). This suggests that the global
Parisian population exposure is reasonably well modelled for $PM_{2.5}$ using regional-scale modelling with 1 km$^2$ resolution.

For $NO_2$ and eBC, the ESF is about 1.25 in Paris indicating that outdoor population exposure is underestimated by as much
as 25% in Paris urban areas when considering only regional-scale concentrations. For PN, the ESF is slightly lower (1.12) in
Paris. In heavy-traffic areas, such as in cells that include the very busy Paris ring road, the ESF varies from 1.27 for $NO_2$ and
1.27 for eBC. The ESF is 1.05 for $PM_{2.5}$ along the ring road, indicating that the population exposure is not well represented by
regional-scale concentrations for people living next to very busy streets.

**Table 5.** Average PWC and ESF for $NO_2$, eBC, $PM_{2.5}$, and PN over the period of simulation in Paris and Paris Ring Road. PWC is in $\mu$g
m$^{-3}$ for $NO_2$, BC, $PM_{2.5}$, and in # cm$^{-3}$ for PN.

|  |  | PWC | | ESF | |
|---|---|---|---|---|---|
|  |  | Paris | Paris Ring road | Paris | Paris Ring Road |
|  | $NO_2$ | 25.34 | 30.14 | 1.26 | 1.27 |
| REF | eBC | 0.57 | 0.66 | 1.24 | 1.27 |
|  | $PM_{2.5}$ | 7.12 | 7.57 | 1.04 | 1.05 |
|  | PN | 10963 | 12816 | 1.12 | 1.13 |
|  | $NO_2$ | 25.34 | 30.14 | 1.26 | 1.27 |
| SEN | eBC | 0.52 | 0.59 | 1.22 | 1.25 |
|  | $PM_{2.5}$ | 6.81 | 7.15 | 1.02 | 1.03 |
|  | PN | 10899 | 12718 | 1.12 | 1.13 |





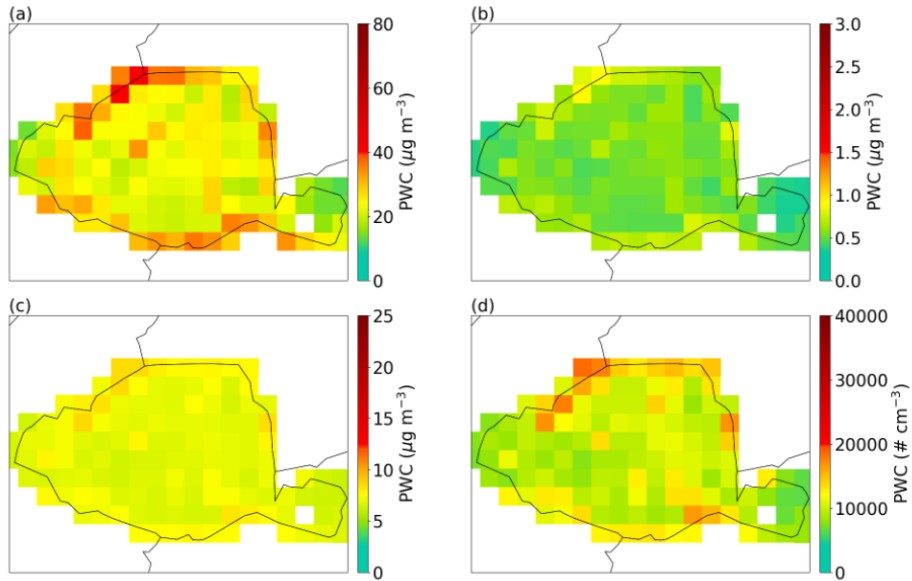

**Figure 10.** PWC of (a) $NO_2$, (b) eBC, (c) $PM_{2.5}$, and (b) PN in the REF simulation.

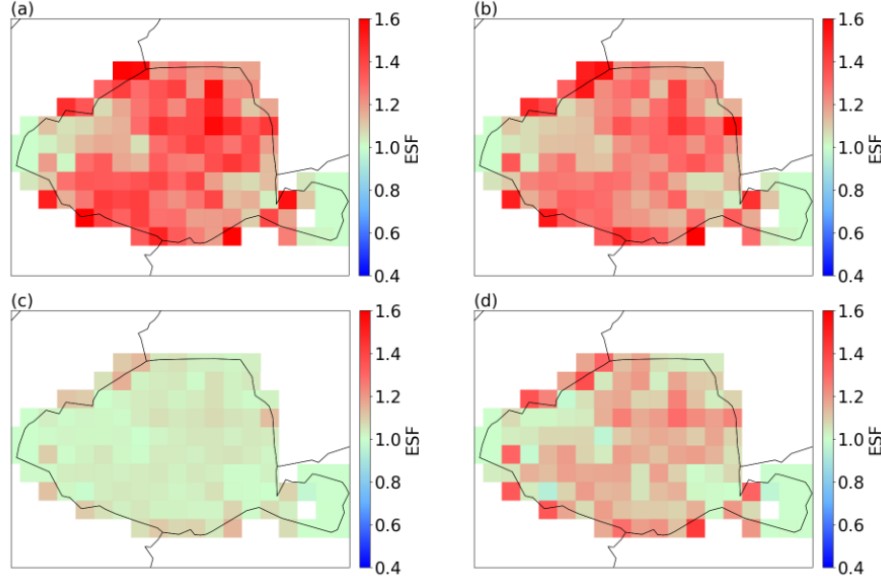

**Figure 11.** ESF of (a) $NO_2$, (b) eBC, (c) $PM_{2.5}$, and (b) PN in the REF simulation.



## 5 Conclusion

This study focuses on modelling pollutants of interest to health (NO$_2$, BC, PM$_{2.5}$, and PN) and with high concentrations in
urban areas, so as to represent the population's exposure to outdoor concentrations as accurately as possible. To do so, a multi-
scale simulation is performed over Greater Paris and down to the street scale over Paris during the summer of 2022 using
WRF-CHIMERE/MUNICH/SSH-aerosol.

In the regional-scale modelling over Greater Paris, the use of two emission inventories was compared: the top-down inventory
EMEP and the bottom-up inventory Airparif with correction of the traffic flow using counting loops. The concentrations were
evaluated compared to measurements using strict and less strict criteria from the literature. For regional-scale urban background
concentrations, strict criteria are met for NO$_2$, PM$_{2.5}$, eBC and PN, although the statistics are better for NO$_2$, eBC and PN using
the bottom-up inventory than the top-down one. At the street-scale, only the bottom-up inventory is used as it provides traffic
emissions per street segments. By comparisons to observations at background and traffic station, the strict criteria are met for
NO$_2$, PM$_{2.5}$, and eBC. As observed, the number of particles is higher on busy streets than in the urban background by a factor
of between 1.8 and 3.

The concentrations of NO$_2$, eBC, PM$_{2.5}$, and PN in Greater Paris are high in streets, particularly along the Paris Ring
Road. Using the top-down inventory, the concentrations tend to be lower in streets than those simulated using the bottom-up
inventory, especially for NO$_2$ concentrations, resulting in less urban heterogeneities. Two estimations of the size distribution
of non-exhaust emissions were compared. The impact on eBC and PM$_{2.5}$ is relatively low at the regional scale, by at most a
few percents. However, it is higher in heavy-traffic streets, leading to an average impact on the concentrations of eBC, PM$_{2.5}$,
and PN concentrations by 9.5%, 6.6%, and 0.7%, respectively.

Population exposure to outdoor concentrations is estimated for Paris by cross-referencing the MAJIC database, which gives
the number of inhabitants in each building, with the multi-scale simulations. An exposure scaling factor (ESF) is then calculated
to estimate the error made by using the 1 km$^2$ resolved regional-scale concentrations to approximate outdoor concentrations.
The ESF is the highest for eBC, followed by NO$_2$, PN, and finally PM$_{2.5}$. The average ESF in Paris is higher than 1 for
all pollutants, which indicates that the Parisian population exposure is under-estimated using regional-scale concentrations.
Although this under-estimation is low for PM$_{2.5}$, with an ESF of 1.04, it is very high for NO$_2$ (1.26), eBC (1.24 in REF and
1.22 in SEN), and PN (1.12). This shows that urban heterogeneities are important to be considered in order to realistically
estimate the population exposure to NO$_2$, eBC, and PN.

Multi-scale simulations using bottom-up traffic emissions provide innovative and detailed spatially-resolved air-quality in-
formation in urban areas. In particular, the ESF may be used to refine the evaluation of population exposure in urban areas
employing regional-scale models. Further investigation is needed to assess the concentrations and population exposure scaling
factors for different seasons and different cities.



*Code and data availability.* The CHIMERE/MUNICH/SSH-aerosol chain used here is available online at https://doi.org/10.5281/zenodo.12639507
(Park et al., 2024).

Hourly $NO_2$ and $PM_{2.5}$ concentrations measured at the Airparif' stations are available on the Airparif's Open Data Portal: https://data-airparif-asso.opendata.arcgis.com. Regional emissions inventory, as well as hourly BC and PN data for Paris Châtelet Les Halles, BP-EST and HAUSS stations can be available on request. $NO_2$ and BC hourly measurements at the HdV station are available on the Aeris data center (https://across.aeris-data.fr/catalogue/). SMPS measurements at SIRTA during the ACROSS campaign can be found on ACROSS data
portal https://across.aeris-data.fr. Other measurements at SIRTA are performed within the framework of ACTRIS and can be found on EBAS https://ebas.nilu.no. Level 2 datasets used in the present study from the ACROSS field campaign for the PRG sites are available on the AERIS datacenter (https://across.aeris-data.fr/catalogue/). The SMPS data obtained at PRG by the SMPS are available using the following DOI 10.25326/658 on the AERIS database. This dataset has the corresponding citation KAMMER, J., SHAHIN, M., D'Anna, B. & Temime-Roussel, B. (2024). ACROSS_LCE_PRG_SMPS_5 min_L2. [dataset]. Aeris. https://doi.org/10.25326/658 The eBC data obtained at PRG
by the AE33 are available using the following DOI 10.25326/575 on the AERIS database (https://across.aeris-data.fr/catalogue/) This dataset has the corresponding citation Di Antonio, L., Di Biagio, C. & Gratien, A. (2023). ACROSS_LISA_PRG_AETH-eBC_PM1_1-Min_L2. [dataset]. Aeris. https://doi.org/10.25326/575.

*Author contributions.* SJP and LL prepared the input data for the CHIMERE/MUNICH model, LL and MV were responsible for the CHIMERE setup and LL performed the CHIMERE simulations. SJP performed the MUNICH simulations. SJP, LL, KS performed the
formal analysis. JV, FD and OS provided the regional and traffic emission inventory. OJ determined the formula to discretize the emission size distribution. SJP and YK were responsible for computing the population exposure. SJP and LL conducted the visualization. The experimental data were provided by AB and VG for the Airparif sites; by AG, CDB, LDA, BDA and JK for the PRG site; by JEP for SIRTA; by JEP and OF for the HdV site. KS was responsible for the conceptualization, funding acquisition and supervision. KS, SJP and LL wrote the original draft, and all authors reviewed it.

*Competing interests.* The authors have no competing interests to declare.

*Acknowledgements.* This project has also received funding from the European Union's Horizon 2020 research and innovation program under grant agreement No 101036245 (RI-URBANS), and from the French Research project ENZU (Evolution of the Number of Particles in Urban areas) under the program AQACIA/Ademe. BC measurements at the traffic station HdV benefited from the French state aid managed by the ANR under the "Investissements d'avenir" program (ANR-11-IDEX-0004-17-EURE-0006) with support from IPSL/Composair. The
measurements at the PRG site have been supported by the ACROSS project. The ACROSS project has received funding from the French National Research Agency (ANR) under the investment program integrated into France 2030 (grant no. ANR-17-MPGA-0002), and it has been supported by the French National program LEFE (Les Enveloppes Fluides et l'Environnement) of the CNRS/INSU (Centre National de la Recherche Scientifique/Institut National des Sciences de L'Univers). The authors also gratefully acknowledge CNRS-INSU for supporting measurements performed at the SI-SIRTA, and those within the long-term monitoring aerosol program SNO-CLAP, both of which are





components of the ACTRIS French Research Infrastructure, and whose data is hosted at the AERIS data center (https://www.aeris-data.fr/)
This project was provided with computer and storage resources by GENCI at TGCC thanks to the grant A0150114641 on the supercomputer
Joliot Curie's the ROME partition. The authors would like to thank Laurent Letinois from INERIS (France) for providing the MAJIC
database. The authors would like to thank Alice Maison for discussions on the computation of the Monin-Obukhov length; Matthieu Riva,
Rulan Verma and Sebastien Perrier from IRCELYON for nano-SMPS measurements at the PRG site; Shravan Deshmukh and Laurent Poulain
for SMPS measurements at the SIRTA site.

## Appendix A:  Size-discretisation of emissions

### A1    Partitioning into two sections by conserving both mass and number

Let $M$, $N$ be the mass and number concentration of particles contained in a section spanning from diameters $d_-$ to $d_+$. We assume that $M$ and $N$ are related by

$$M = N\frac{\pi}{6}\rho\bar{d}^3 \tag{A1}$$

with $\bar{d} = (d_- d_+)^{\frac{1}{2}}$ the geometric mean diameter of the section boundaries.

The section defined by boundaries $d_-$ and $d_+$ is partitioned in two new sections. For that purpose, we define a new diameter $d_\mathrm{m}$ such that $d_- < d_\mathrm{m} < d_+$. The first section spans from $d_-$ to $d_\mathrm{m}$ with mass and number concentrations $M_1$ and $N_1$, while the second section spans from $d_\mathrm{m}$ to $d_+$ with mass and number concentrations $M_2$ and $N_2$.

Assuming both mass and number conservation, implyies

$$M = M_1 + M_2 \tag{A2}$$

$$N = N_1 + N_2 \tag{A3}$$

Furthermore, we enforce a relation similar to (A1) for each section

$$M_1 = N_1\frac{\pi}{6}\rho(d_- d_\mathrm{m})^{\frac{3}{2}} \tag{A4}$$

$$M_2 = N_2\frac{\pi}{6}\rho(d_\mathrm{m} d_+)^{\frac{3}{2}} \tag{A5}$$





Since we have introduced 5 new variables ($d_{\mathrm{m}}$, $M_1$, $M_2$, $N_1$, $N_2$) and 4 new equations, we should therefore be able to
express mass and number concentrations in the partitions as a function of previous variables and $d_{\mathrm{m}}$.

$$
\begin{aligned}
M_1 &= N_1 \frac{\pi}{6} \rho (d_- d_{\mathrm{m}})^{\frac{3}{2}} \\
&= (N - N_2) \frac{\pi}{6} \rho (d_- d_{\mathrm{m}})^{\frac{3}{2}} \\
&= M \sqrt{\frac{d_{\mathrm{m}}^3}{d_+^3}} - M_2 \sqrt{\frac{d_-^3}{d_+^3}} \\
&= M \sqrt{\frac{d_{\mathrm{m}}^3}{d_+^3}} - (M - M_1) \sqrt{\frac{d_-^3}{d_+^3}}
\end{aligned}
$$

Therefore

$$
M_1 = M \frac{d_{\mathrm{m}}^{3/2} - d_-^{3/2}}{d_+^{3/2} - d_-^{3/2}} \tag{A6}
$$

and similarly

$$
M_2 = M \frac{d_+^{3/2} - d_{\mathrm{m}}^{3/2}}{d_+^{3/2} - d_-^{3/2}} \tag{A7}
$$

### A2 Partitioning into $n$ sections

We wish to upsample our initial section into $n$ sections, defined by the boundaries $\{d_i\}_{i \in (0,n)}$ such that $d_{i-1} < d_i$ for all
$i \in (1,n)$. A more general relation for partitioning into $n$ sections can be deduced by noticing that any subsection can be
obtained by performing recursively at most two partitions of the whole diameter range.

For the trivial case of the lowest and highest diameter sections, we only need one partitioning operation to deduce their
concentrations :

$$
M_1 = M \frac{d_1^{3/2} - d_0^{3/2}}{d_n^{3/2} - d_0^{3/2}}
$$

$$
M_n = M \frac{d_n^{3/2} - d_{n-1}^{3/2}}{d_n^{3/2} - d_0^{3/2}}
$$

For inner sections such that $1 < i < n$, two successive partitioning operations are necessary. Indeed, by splitting the full
diameter range on the lowest diameter of our section of interest and successively splitting the new section on the highest
diameter of our section of interest, we have managed to create a section with the boundaries we wished to enforce. The section
composition can therefore be inferred.



$$M_i = \left( M \, \frac{d_n^{3/2} - d_{i-1}^{3/2}}{d_n^{3/2} - d_0^{3/2}} \right) \times \frac{d_i^{3/2} - d_{i-1}^{3/2}}{d_n^{3/2} - d_{i-1}^{3/2}}$$

$$M_i = M \frac{d_i^{3/2} - d_{i-1}^{3/2}}{d_n^{3/2} - d_0^{3/2}} \tag{A8}$$

Furthermore, one can easily check that equation (A8) also holds for $i = 1$ and $i = n$.



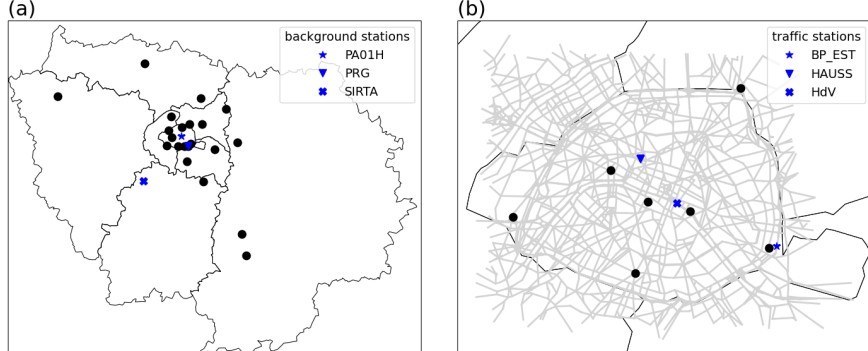

**Figure B1.** (a) Urban background and (b) traffic stations. Black dots represent the other Airparif stations used in this study.

## A3 Partitioning into the 10 sections used here

With the size sections employed in this study and the methodology defined above, the ratios obtained to split emissions in each size section are indicated in Table A1.

**Table A1.** Ratios employed in PM emissions to split the particle classes into the different size sections

| Size section | dmin [$\mu$m] | dmax [$\mu$m] | PM class | Ratio [%] |
|:---:|:---:|:---:|:---:|:---:|
| 1 | 0.01 | 0.0199 | $PM_{0.1}$ | 2.91 |
| 2 | 0.0199 | 0.0398 | $PM_{0.1}$ | 8.27 |
| 3 | 0.0398 | 0.0794 | $PM_{0.1}$ | 23.24 |
| 4 | 0.0794 | 0.1585 | $PM_{0.1}$ | 65.58 |
| 5 | 0.1585 | 0.316 | $PM_1$ | 26.14 |
| 6 | 0.316 | 0.631 | $PM_1$ | 73.86 |
| 7 | 0.631 | 1.256 | $PM_{2.5}$ | 26.26 |
| 8 | 1.256 | 2.5 | $PM_{2.5}$ | 73.74 |
| 9 | 2.5 | 5 | $PM_{10}$ | 26.12 |
| 10 | 5 | 10 | $PM_{10}$ | 73.88 |

**Appendix B: Supplementary figures**





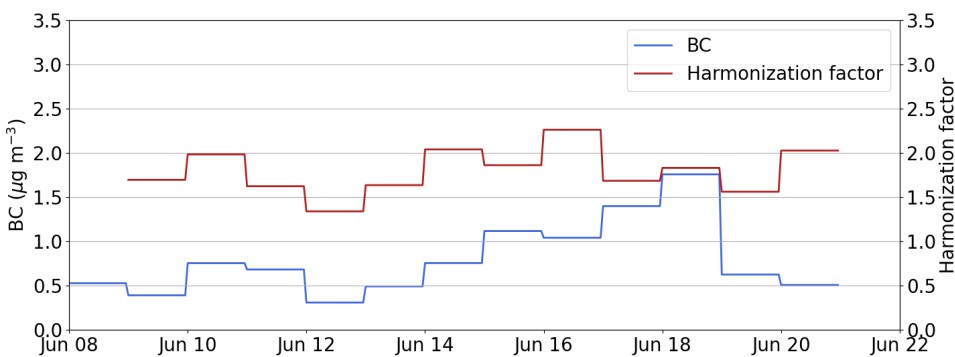

**Figure B2.** Time series of BC mass and harmonization factor at the PA01H station.

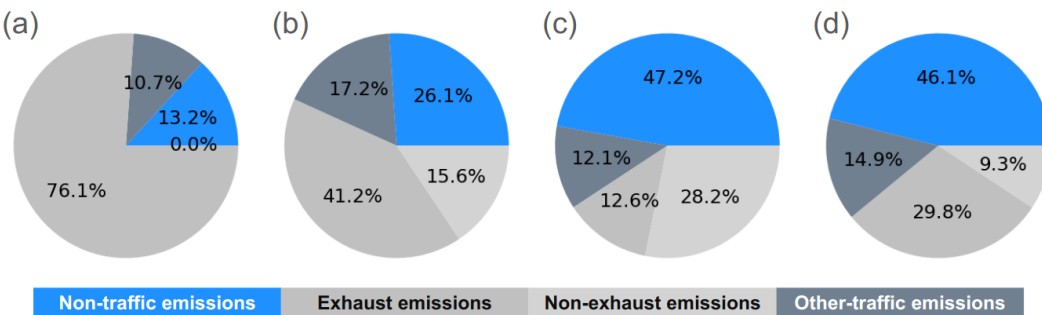

**Figure B3.** Distribution between vehicular traffic (exhaust and non-exhaust), other traffic and non-traffic emissions for NOx (a), BC (b), PM$_{2.5}$ (c), and PN (d) in Paris.



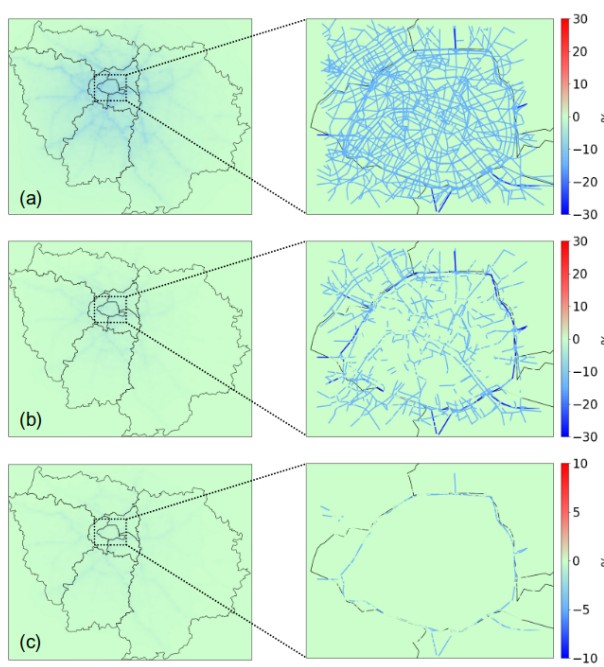

**Figure B4.** Relative difference (%) in (a) eBC, (b) PM$_{2.5}$, and (c) PN concentrations between the SEN and REF simulations.

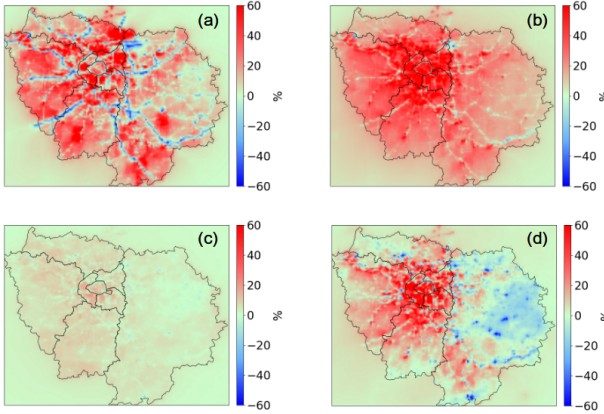

**Figure B5.** Relative differences (%) in (a) NO$_2$, (b) eBC, (b) PM$_{2.5}$, and (d) PN concentrations between the EMEP and REF simulations.



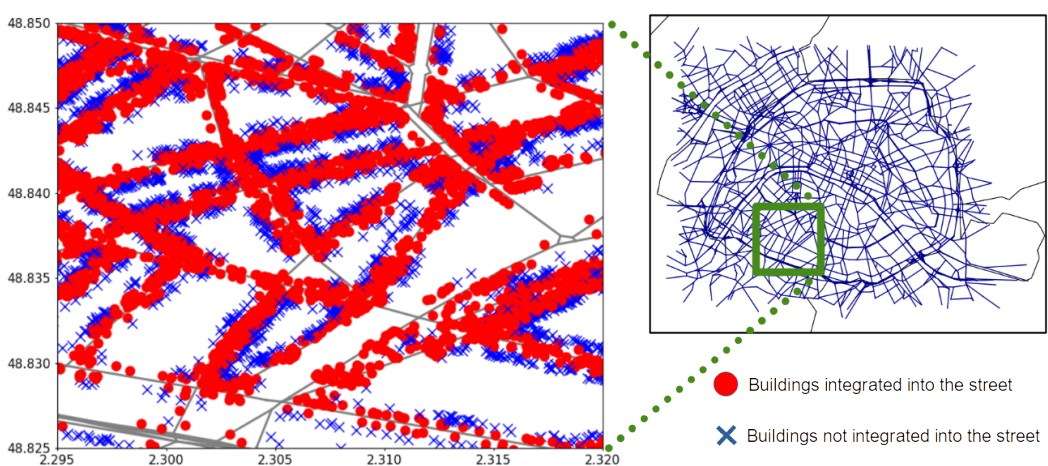

**Figure B6.** The distribution of buildings according to their integration into the street. The red circles indicate buildings integrated into the street, while the blue crosses indicate buildings that are not integrated into the street.



## Appendix C: Statistical parameters

### C1  Definitions

$$FB = 2 \cdot \left( \frac{\overline{O} - \overline{S}}{\overline{O} + \overline{S}} \right)$$

$$MG = exp[\overline{ln(O)} - \overline{ln(S)}]$$

$$NMSE = \sqrt{\frac{\overline{(O-S)^2}}{\overline{O \cdot S}}}$$

$$VG = exp[\overline{(ln(O) - \ln(S))^2}]$$

$$NAD = \frac{|\overline{O-S}|}{\overline{(O+S)}}$$

$$FAC2 = \text{fraction of data that satisfy: } 0.5 \le \frac{S}{O} \le 2.0$$

$$NMB = \frac{\overline{(S-O)}}{\overline{O}}$$

$$NME = \frac{|\overline{(S-O)}|}{\overline{O}}$$

where O and S represent the observed and simulated concentrations, respectively.



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
