# Peer review of "Population exposure to outdoor NO2, black carbon, ultrafine and fine particles over Paris with multi-scale modelling down to the street scale"

_EGUsphere, 2024_

## Author Response (AR2)

**Population exposure to outdoor NO₂, black carbon, ultrafine and fine particles over Paris with multi-scale modelling down to the street scale**

Soo-Jin Park[1], Lya Lugon[1], Oscar Jacquot[1], Youngseob Kim[1], Alexia Baudic[7], Barbara D'Anna[8], Ludovico Di Antonio[3], Claudia Di Biagio[4], Fabrice Dugay[7], Olivier Favez[5], Véronique Ghersi[7], Aline Gratien[4], Julien Kammer[8], Jean-Eudes Petit[6], Olivier Sanchez[7], Myrto Valari[2], Jérémy Vigneron[7], and Karine Sartelet[1]

[1]CEREA, Ecole des Ponts, EDF R & D, Institut Polytechnique de Paris, IPSL, Marne-la-vallée, France
[2]LMD/IPSL, École Polytechnique, Université Paris Saclay, ENS, PSL Research University, Sorbonne Universités, UPMC Univ, France
[3]Université Paris Est Créteil and Univ. Paris Cité, CNRS, LISA, F-94010 Créteil, France
[4]Université Paris Cité and Univ. Paris Est Créteil, CNRS, LISA, F-75013 Paris, France
[5]INERIS, 60550 Verneuil en Halatte, France
[6]Laboratoire des Sciences du Climat et l'Environnement, CEA/Orme des Merisiers, 91191 Gif-sur-Yvette, France
[7]Airparif, 75004, Paris, France
[8]Aix Marseille Univ, CNRS, LCE, Marseille, France

**Reply to reviewer 2**

*The authors need to modify the model-to-data comparison to the model-to-observation comparison.*
**Our reply:** Modified.

5

*- Lines 157 & 158: The expression of 'hours×processors' is inappropriate. Describe it specifically.*

**Our reply:** We have changed expressions of 'hours×processors' to "processor hours", which commonly used in journal papers, and additionally included the specific time and number of processors in parentheses.

10 "Using a one-way coupling approach, the regional-scale and local-scale simulations are performed sequentially. For the regional scale, the two-month simulation using WRF-CHIMERE models requires approximately 11520 processor hours (192 processors over 60 hours). The local scale simulations are less expensive, and the two-month simulation with the MUNICH model requires around 7680 processor hours (64 processors over 120 hours) to simulate the street concentrations in the Parisian street-network composed of 4655 streets."

15

*- Line 180: Title: Review and correct the title of PN emission if incorrect. PNC emission (?).*

**Our reply:** The title of PN emission is correct, because emissions are different from concentrations, and in PNC, the "C" stands for concentrations. The term "PNC emission" was modified to "PN emission".

20